# Docosahexaenoic Acid (DHA) Bioavailability in Humans after Oral Intake of DHA-Containing Triacylglycerol or the Structured Phospholipid AceDoPC^®^

**DOI:** 10.3390/nu12010251

**Published:** 2020-01-18

**Authors:** Mayssa Hachem, Houda Nacir, Madeleine Picq, Mounir Belkouch, Nathalie Bernoud-Hubac, Anthony Windust, Laure Meiller, Valerie Sauvinet, Nathalie Feugier, Stephanie Lambert-Porcheron, Martine Laville, Michel Lagarde

**Affiliations:** 1Univ-Lyon, CarMeN Laboratory, Inserm UMR 1060, Inra UMR 1397, IMBL, INSA-Lyon, 69100 Villeurbanne, France; mhachem@amityuniversity.ae (M.H.); houda.nacir@yahoo.fr (H.N.); madeleine.picq@insa-lyon.fr (M.P.); mbelkouch@yahoo.fr (M.B.); nathalie.bernoud-hubac@insa-lyon.fr (N.B.-H.); laure.meiller@chu-lyon.fr (L.M.); valerie.sauvinet@chu-lyon.fr (V.S.); martine.laville@chu-lyon.fr (M.L.); 2National Research Council Canada, Ottawa, ON K1A 0R6, Canada; ajwindust@gmail.com; 3Hospices Civils de Lyon, Groupement Hospitalier Sud, 69310 Pierre-Bénite, France; nathalie.feugier@chu-lyon.fr (N.F.); stephanie.lambert-porcheron@chu-lyon.fr (S.L.-P.); 4CRNH Rhône-Alpes, CENS, 69310 Pierre-Bénite, France

**Keywords:** plasma phospholipids, brain, gas chromatography combustion isotope ratio mass spectrometry

## Abstract

AceDoPC^®^ is a structured glycerophospholipid that targets the brain with docosahexaenoic acid (DHA) and is neuroprotective in the experimental ischemic stroke. AceDoPC^®^ is a stabilized form of the physiological 2-DHA-LysoPC with an acetyl group at the *sn1* position; preventing the migration of DHA from the *sn2* to *sn1* position. In this study we aimed to know the bioavailability of ^13^C-labeled DHA after oral intake of a single dose of ^13^C-AceDoPC^®^, in comparison with ^13^C-DHA in triglycerides (TAG), using gas chromatography/combustion/isotope ratio mass spectrometry (GC/C/IRMS) to assess the ^13^C enrichment of DHA-containing lipids. ^13^C-DHA enrichment in plasma phospholipids was significantly higher after intake of AceDoPC^®^ compared with TAG-DHA, peaking after 24 h in both cases. In red cells, ^13^C-DHA enrichment in choline phospholipids was comparable from both sources of DHA, with a maximum after 72 h, whereas the ^13^C-DHA enrichment in ethanolamine phospholipids was higher from AceDoPC^®^ compared to TAG-DHA, and continued to increase after 144 h. Overall, our study indicates that DHA from AceDoPC^®^ is more efficient than from TAG-DHA for a sustained accumulation in red cell ethanolamine phospholipids, which has been associated with increased brain accretion.

## 1. Introduction

Docosahexaenoic acid (DHA)/22:6n-3 is the long-chain polyunsaturated n-3 fatty acid (n-3 PUFA) which specifically accumulates into the brain where it plays a crucial role for the development and function [1]. Indeed, DHA is required for brain development from fetus to adult, and for cognition and visual acuity [2]. It is involved in brain development such as synaptogenesis, neurogenesis, neuronal differentiation [3], and maintenance of membrane fluidity [4]. In addition, brain DHA content is altered in neurodegenerative diseases, particularly in Alzheimer’s disease [5,6]. DHA synthesis from oral intake of its essential precursor α-linolenic acid (18:3n-3) is only 1% in men and 1–3% in women [7]. Due to this low level of synthesis, dietary pre-formed DHA is the preferred source of DHA to improve brain DHA accretion.

It is known that DHA comes from blood plasma through crossing the blood-brain barrier (BBB), both as a non-esterified fatty acid and as esterified in lyso-phosphatidylcholine (lysoPC). However, several studies have shown that, although a slower uptake of DHA was observed from DHA-containing lysoPC, a stronger DHA accumulation into the brain was found on a long term [8,9], whereas the brain uptake of DHA is faster but more limited from non-esterified DHA [10,11]. Indeed, DHA esterified at the *sn2* position of lysoPC more efficiently crosses a re-constituted BBB than non-esterified DHA [12]. This has been corroborated by more recent studies showing that BBB expresses the specific binding protein Mfsd2a to facilitate this transfer [13], and that an oral intake of DHA-containing lysoPC is more efficient than non-esterified DHA to increase the brain DHA content and the memory [9]. It is noteworthy that the brain uptake of other unsaturated fatty acids, especially arachidonic acid, which is the second most abundant after DHA, also is most efficient when esterified in lysoPC [14].

However, DHA esterified at the *sn2* position of lysoPC, supposed to be its physiological position within phosphoglycerides, rapidly migrates to the *sn1* position [15]. We then structured such a lysoPC to prevent this migration, by acetylating the *sn1* position [16]. The resulting structured phospholipid: 1-acetyl, 2-docosahexaenoyl-glycerophosphocholine has been named AceDoPC^®^. This compound affords more neuroprotection in a post-ischemic stroke model than does non-esterified DHA [17], and DHA incorporation into brain tissues is greater with AceDoPC than with equimolar concentrations of non-esterified DHA or of PC DHA [18]. On the other hand, additional findings showed that AceDoPC^®^ inhibits lipopolysaccharide-induced neuro-inflammation in microglial cells through interleukin-6 signaling [19], and activates neurogenesis, but not astrogenesis, from nerve stem cells [20]. In light of this potential, it was of interest to investigate the bioavailability of DHA in humans from oral intake of AceDoPC^®^ compared to a usual source such as TAG-DHA.

## 2. Materials and Methods

### 2.1. Human Volunteers

Three healthy men between 60 to 70 years old, with no cognitive defects, were selected. Their body mass index was between 20 to 30 Kg/m^2^. In whole blood of fasting subjects, glycaemia was less than 7 mmol/L, total cholesterol was less or equal to 7 mmol/L, triglycerides level was less than 1.7 mmol/L, and hemoglobin was more than 130 g/L. All subjects gave their informed consent for inclusion before participating in the study.

### 2.2. Experimental Procedure

The research was conducted at the clinical research center from Rhône-Alpes Research Center for Human Nutrition (CRNH-RA) under the Clinical Trial Number NCT02168738. The Sponsor was the Hospices Civils de Lyon, and the agreement was obtained from the legal authorities, after approvement by the local ethics committee. This was a double-blind randomized study with a crossover use of two sources: ^13^C-AceDoPC and TAG-^13^C-DHA. Each source contained 50 mg of ^13^C-DHA that was ingested after 12 h of fasting, as an alcoholic solution on a piece of bread. The blood sampling (36 mL in total) started before the DHA ingestion (T0), and after 1 (T1), 3 (T3), 6 (T6), 24 (d1), 72 (d3), and 144 (d6) hours, of the oral intake. A wash-out of 120 days between each source was applied to avoid or minimize ^13^C labeling of red blood cells from the first intake. 

### 2.3. Synthesis of Labeled DHA Sources

#### 2.3.1. ^13^C-AceDoPC Synthesis

The synthesis of labeled AceDoPC with ^13^C-DHA was similar to that of [^14^C]-AceDoPC as previously described [18], using a DHA ester form, U^13^C-DHA methyl ester (provided by Dr. Anthony Windust’s lab) uniformly labeled with carbon 13. The purity of the synthesized product was checked by High Performance Liquid Chromatography (HPLC), and ^13^C Nuclear Magnetic Resonance (NMR). Finally, DHA was analyzed by Gas Chromatography (GC) and the ^13^C enrichment was checked by GC/mass spectrometry (MS), showing uniformly labeled DHA. 

#### 2.3.2. Production of ^13^C-DHA-Containing Triacylglycerol

^13^C-TAG was produced by growing a microalgae (*Crypthecodinium cohnii)* strain (ATCC-30772 American Type Culture Collection) in a defined medium by Nestlé industry (Tours, France). All the products came from Sigma or Riedel-deHaën. The microalgae were developed during several days in a specific medium with salt and water in dark conditions. Microalgae were transferred in a starting medium, at pH 6.5 and 27 °C, containing 1-^13^C-acetate. The biomass was collected after 20 days of microalgae growing.

^13^C-DHA-TAG was harvested from the cells by centrifugation. The biomass was then freeze-dried, and lipids were extracted with hexane/isopropanol/water 5:5:1 (*v/v/v*). Lipids were separated by HPLC, using a normal phase silica column (21 × 250 mm; 10 µm). The solvent system used was a gradient consisting of hexane/2-propanol (55:4, *v/v*) (solvent A) to hexane/isopropanol/water (60/120/20, *v/v/v*) (solvent B) with a flow rate of 30 mL/min. Neutral lipids were eluted and collected between 0 and 8 min. The fatty acid composition of TAG was obtained by GC showing that DHA represented 20% of total fatty acids in TAG. Analysis of proton spectrum of ^13^C-TAG showed high purity of the samples according to NMR Bruker 400 MHz experiment (10 mg of samples prepared in 0.75 mL CDCL_3_). The GC/MS analysis confirmed eleven ^13^C within one molecule of DHA. The non-toxicity for human administration was checked by pharmacists from Hospices Civils de Lyon.

### 2.4. Blood Red Cells and Plasma Preparation

Blood samples were collected in tubes containing citrate-dextrose and the red cells and platelet-poor-plasma were obtained as described previously [21]. Plastic tubes containing whole blood were centrifuged at 200× *g* for 15 min at 4 °C to obtain platelet-rich plasma (PRP) and the red blood cells pellet. This PRP was acidified to pH 6.4 with citric acid to prevent platelet activation and centrifuged at 900× *g* for 12 min at 4 °C to pellet platelets and get the platelet poor plasma (PPP) in the supernatant. This PPP was then centrifuged at 2000× *g* for 10 min at 4 °C to eliminate the remaining platelets, and the plasma supernatant was added with 5 × 10^−5^ M butyl hydroxyl-toluene (BHT) as an antioxidant, and frozen at −80 °C until purification of lipids. Blood red cells/erythrocytes, from the first 200× *g* pellet, were diluted in Tyrode-HEPES buffer and centrifuged at 100× *g* for 10 min at 4 °C to remove contaminating white cells. After removing the supernatant, red blood cells were diluted with 9% NaCl and centrifuged at 2000× *g* for 10 min at 4 °C. This was repeated once and Tyrode-HEPES was added to red cells, in presence of 5 × 10^−5^ M BHT as an antioxidant, and stored at −80 °C.

### 2.5. Extraction and Separation of Lipids from Red Cells and Plasma

Total lipids were extracted from red cells and plasma according to Bligh and Dyer. Internal standards (1,2-diheptadecanoyl-glycerophosphocholine and 1,2-diheptadecanoyl-glyceroethanolamine) were added to samples before lipid extraction for quantification. Lipid classes were separated by thin-layer chromatography (TLC) with first diethylether/methanol (90:10, *v/v*) as the mobile phase to elute nonpolar lipids. Then, total phospholipids were extracted from the origin, or phosphatidylethanolamine (PE) and phosphatidylcholine (PC) were separated with chloroform/methanol/water (63:27:4, *v/v/v*). Total phospholipids or PC and PE were scraped off the plate and treated for 90 min with boron trifluoride in methanol/toluene (50:50, *v/v*) to obtain fatty acid methyl esters.

### 2.6. Quantification of ^13^C by Gas Chromatography Combustion-Isotope Ratio Mass Spectrometry (GC-C-IRMS)

The ^13^C enrichment was measured by GC-C-IRMS (gas chromatography–combustion isotope ratio mass spectrometry) (Isoprime; Elementar UK Ltd, Cheade SK8 6PT, UK.) [22]. The sample was injected into a gas chromatograph (model GC6890, Agilent Technologies, Palo Alto, CA, United States) equipped with a fused-silica column (SP2380, 30 m × 0.25 mm × 0.20 µm film thickness, Supelco). Helium was used as the carrier gas. Injection (1 µL) was performed in split less mode at 250 °C, with a split less time of 1min. DHA was separated at constant flow (1.2 mL min^–1^) with the following oven program: (a) 100 °C for 1 min; (b) increase at a rate of 25 °C min^–1^ to 175 °C; (c) hold at 175 °C for 5.3 min; (d) increase at a rate of 4 °C min^−1^ to 250 °C. The GC effluent was diverted to a flame ionization detector until the elution of the target peak. The effluent from the gas chromatograph was then switched into a copper oxide furnace maintained at 850 °C which produced CO_2_ and water from the sample. Water and CO_2_ passed through Nafion tubing, where water was removed, while CO_2_ was transferred to the IRMS instrument *via* an open-split interface. Before and after the CO_2_ peak arising from sample combustion, a reference CO_2_ gas calibrated against PDB was sequentially injected in the IRMS instrument for 30 s, where PDB is the Pee Dee Belemnite international standard ((^13^C/^12^C)_PDB_ = 0.0112372). The different isotopomers were collected onto three different collectors at mass-to-charge ratio (*m/z*) 44 (main ion: ^12^C^16^O^16^O), 45 (^13^C^16^O^16^O, ^12^C^16^O^17^O), and 46 (^12^C^17^O^17^O, ^12^C^16^O^18^O, ^13^C^16^O^17^O). Ions at *m/z* 44, 45, and 46 were continuously recorded until the return of the 44 signal to the baseline value. Isotopomers at *m/z* 44 and 45 were measured, leading to the ^13^C/^12^C ratio. The ^13^CO_2_/^12^CO_2_ ratios of samples were expressed as δ^13^C ‰ relative to PDB:δC 13sample =[(C 13C 12)sample(C 13C 12)PDB−1]×1000

To take into account the difference of labelling of the DHA between ^13^C-AceDoPC (22 ^13^C atoms) and ^13^C-TAG (11 ^13^C atoms) results were finally expressed in mole percent excess (MPE) calculated from the Atom%:At% = [100 × (^13^C/^12^C) _PDB_ × (0.001 × δ ^13^C_sample_ + 1)]/[1 + (^13^C/^12^C) _PDB_ × (0.001 × δ ^13^C_sample_ + 1)

### 2.7. Measures of Quality Control

A quality control standard including DHA compound (Mix 37 from Supelco) was injected before and after each batch of analyses. Less than 4.8% variation was observed for ^13^C-DHA enrichment during the study period.

### 2.8. Statistical Analysis

Results are means ± SEM values of *n* = 3. Data with different signs are significantly different at *p* < 0.05, according to the Student *t* test.

## 3. Results

### 3.1. ^13^C-DHA Incorporation into Plasma Phospholipids

After separation of total phospholipids fraction by TLC, FAMEs were analyzed by GC-C-IRMS as described in Materials & Methods. The ^13^ C enrichment in DHA was calculated and reported as the amount of ^13^C-DHA content in the analyzed samples (Figure 1). An increase from both DHA sources (^13^C-AceDoPC and ^13^C-DHA-TAG) started after 3 h of DHA intake, but was substantial after 6 h, and peaked after 24 h. However, the ^13^ C-DHA peak was 2-fold higher from AceDoPC (5386 pmol/mL) compared to TAG-DHA (3247 pmol/mL). After peaking, the ^13^C enrichment in DHA decreased at 72 h from both sources but remained significantly different. This decrease continued until 144 h, but no more significance between the two sources could be seen. The statistical significance could then be observed after 24 h and 72 h.

### 3.2. ^13^C-DHA Incorporation into Red Cells PC and PE

^13^C-DHA appeared in red blood cells (phosphatidylcholines + phosphatidylethanolamines: PC + PE) after 6 h, and increased linearly till 72 h, then plateauing at 144 h, although ^13^C-DHA tended to decrease from TAG-DHA while tending to increase from AceDoPC. (Figure 2A). The ^13^C-DHA appearance into PC exhibited a peak at 72 h and decreased at 144 h (Figure 2B). No significant differences were found for these kinetics for both sources. When studying the accumulation of ^13^C-DHA into PE, it tended to plateauing at 144 h from ^13^C-DHA-TAG whereas it sharply increased at this latter time from ^13^C-AceDoPC, with around twice as much of ^13^C-DHA from AceDoPC (Figure 2C). This strongly suggests that the transfer of DHA over time, which is expected from PC to PE, was more important from AceDoPC than from TAG-DHA.

Overall, Figure 2 shows a transfer over time of DHA from PC to PE, which was predominant when the oral source was AceDoPC compared to TAG-DHA.

## 4. Discussion

It is usually considered that TAG are hydrolyzed at the intestinal level, followed by the absorption of unesterified fatty acids and the remaining 2-acyl-glycerol. Therefore, the DHA bioavailability from TAG-DHA will be affected by its position within the TAG. Considering AceDoPC, DHA being exclusively at the *sn2* position, we might expect its release by phospholipase A_2_, as it occurs from classical phospholipids, except if AceDoPC sufficiently mimicks a lysoPC to be absorbed without hydrolysis. ^13^C-DHA bioavailability from TAG-DHA and AceDoPC should then be different. The results obtained above suggest a higher bioavailability of ^13^ C-DHA from the latter form, with an almost double amount of ^13^C-DHA in plasma phospholipids.

The kinetics of ^13^C-DHA accumulation in red cell phospholipids compared to plasma clearly show a transfer from the latter to the former. In a previous study with ^13^ C-DHA-TAG only, we found there was still increased accumulation in red cell phospholipids after 3 days [23]. Also, in another study looking at the ^13^C-DHA distribution from a single oral intake of phosphatidylcholine (^13^C-DHA-PC) in three human volunteers [24], the last time analysis was after 3 days which did not allow to discriminate between PC and PE accumulation kinetics in red cells. In the current study, the last blood withdrawing was done after 6 days, which reveals a plateau in DHA accumulation in total red cell phospholipids and a different kinetics between PC and PE. Interestingly, ^13^C-DHA in PC peaked after 3 days and started to decrease at 6 days while it still continued to accumulate into PE. Such a difference in the incorporation of ^13^C-DHA within erythrocyte PE is in agreement with the preferential accumulation of DHA into brain PE after dietary intake [25]. Although ^13^C-DHA from TAG-DHA started to plateau after 3 days, it continued to rise from AceDoPC. This suggests that AceDoPC would better promote the brain DHA accretion, as erythrocyte DHA is considered as a marker of brain DHA [26]. This agrees with the observation that AceDoPC, injected i.v. to rats, is efficient to bring DHA to the brain [18], if we speculate that part of this transporter crosses the intestine. Alternatively, it may be considered that AceDoPC might acetylate intestinal targets in an aspirin-like activity [27], then releasing lysoPC-DHA for easily crossing the intestine like other lysoPC from PC-DHA hydrolysis. The recent paper from Sugasini et al. is in favor of this easy crossing [9].

## 5. Conclusions

In summary, oral intake of AceDoPC, even as a single moderate dose (50 mg) would facilitate DHA accumulation in human red cell PE, which is a marker of brain DHA accretion, likely because of a similar structure to lysoPC-DHA. Showing human brain enrichment itself with DHA from dietary AceDoPC would require other approaches, such as brain imaging after intake of appropriately labeled AceDoPC.

## Figures and Tables

**Figure 1 nutrients-12-00251-f001:**
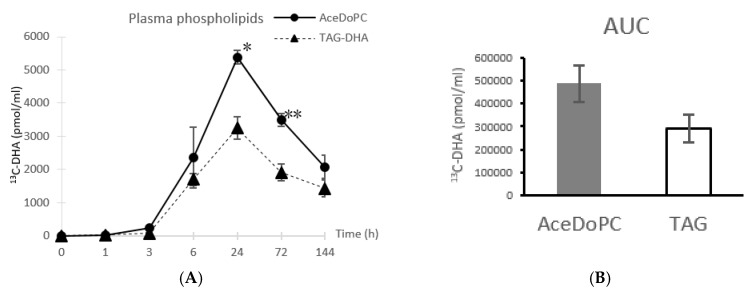
^13^C-DHA in plasma phospholipids from AceDoPC compared to TAG-DHA after 50 mg DHA intake in both esterified forms, at different times post-intake (**A**). Results are expressed in pmol of ^13^C-DHA per mL of plasma, presented as means ± SEM from three values. (**B**) represents area under curves (AUC) from Figure 1A. Stars indicate significant differences within each time, according to the student *t* test.

**Figure 2 nutrients-12-00251-f002:**
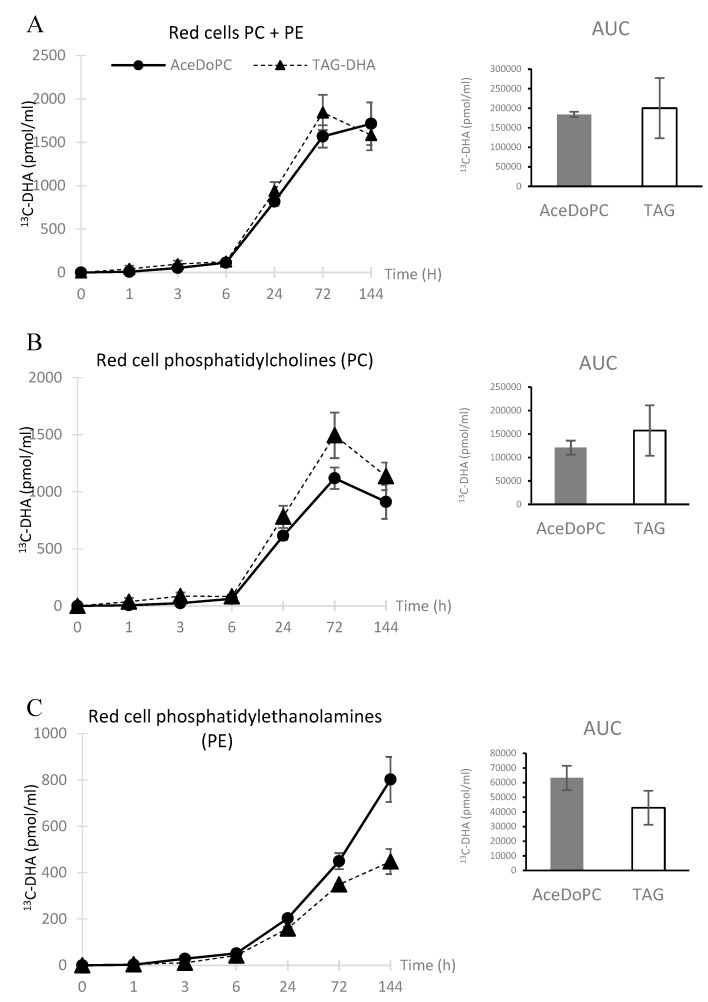
^13^C-DHA in red cell phospholipids after intake of ^13^C-DHA esterified in either AceDoPC or TAG. Fifty milligrams of ^13^C-DHA source were ingested, and blood samples collected at the different times shown in figures. PE & PC were separated as described in Materials & Methods. Results are expressed in pmol of ^13^C-DHA per mL of blood, presented as ± SEM from three values. (**A**) relates to ^13^C-DHA incorporation into red cell PC+PE. (**B**,**C**) relate to separate phospholipids PC and PE, respectively.

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
