# Peer review of "Docosahexaenoic Acid (DHA) Bioavailability in Humans after Oral Intake of DHA-Containing Triacylglycerol or the Structured Phospholipid AceDoPC®"

_nutrients, 2020, doi:10.3390/nu12010251_

Round 1

Reviewer 1 Report

The manuscript by Hachem et al on „Docosahexaenoic acid bioavailability in humans after oral intake of DHA-containing triacylglycerol or structured phospholipid AceDoPC“ is well written and provides some interesting information.

Nevertheless, I would like to suggest some points to consider

I think the description of the synthesis of the used tracers could be very much shortened or even replaced by a reference.

Similarly the description of GC-combustion-IRMS could be shortened, but might be good to add measures describing the analytical performance, measures of quality control.

A paragraph on the statistics and the statistical power of the study should be added. If there was no power calculation in relation to the points described in this manuscript this should be indicted.

There is a good description of the 13C/12C ration and how delta values are calculated, but results are described as concentrations of 13C-DHA. An explanation how these numbers are calculated is missing.

The discussion does not really comment on bioavailability, as suggested by the title and can be seen in Figure 1, but only speculates about a potential mechanism.

On the other hand, I do not find it so clear that the observations in the RBC support a higher brain accretion by AceDoPC than by triglycerides. The curves for PC+PE look very similar. Only the PE curve suggest a higher DHA incorporation from AceDoPC. I think this requires some more discussion.

There is no section on limitations of the study. A point I see is, that all is based on phospholipids. Data on triglycerides and NEFA might help to understand the findings.

What about the diet  of the subjects during the observation period?

Minor points

Line 37: the sentence is unclear, one wonders which  percentage is meant?

Line 169: the sentence is unclear?

In all graphs the distances on the x-axis should be proportional to the numbers: the distance between 0 and 1 should be smaller than the distance between 6 and 24

Author Response

Thanks a lot for your comments. Please find my response in the attachment.

Reviewer 2 Report

I recognize that obtaining the C13-labeled materials is difficult and expensive, but overall, I’d suggest that you be more tentative in your conclusions. This is a study with only a single dose, done in 3 people, followed for only 6 days, without a comprehensive assessment of DHA levels in all blood cell and lipid compartments and without – most importantly - any direct measurements of brain DHA uptake.

For all the incorporation curves, I’d recommend also calculating AUCs and reporting them. It’s unlikely that you can ascribe statistical significance to any of these apparent differences due to the n=3. However, the differences expressed as AUC may not be all that great.

Where is the DHA molecule in the TG? You note that where the DHA ends up after absorption from a TG could differ depending on whether the DHA is liberated or stays with the glycerol at position 2.

You reported plasma phospholipids as a single class. Doing the same for RBC would be helpful.

Is there any evidence that RBC-associated DHA is directly involved with DHA transfer to the brain, or is transfer via plasma-based DHA moieties? If so, it would be more useful to report plasma PL subfraction enrichment than RBC.

L26: bioavailability is a very vague term and should not be in the abstract unmodified especially given the n=3, the single dose, etc. Since you did not calculate AUCs or follow the concentration curves back out to 0, you can’t use the word bioavailability properly. The fact that you’re studying a drug candidate (apparently), it’s best to not ‘over promise’ or exaggerate the findings’ especially in the abstract which is all that most people will read.

L27. “supposed to mimic” could be “has been associated with increased brain accretion.”

L40-41: the difference between “efficiently” and “rapidly” should be explained, and the use of efficiently in L54 is vague. Relative to what?

L56-57: I would suggest, “In light of these findings, it was of interest to investigate...

L61ff: with only 3 people you don’t need ranges or averages; you can list age and BMI for each one.

L66. What is CRNH-RA?

L67. you mean an ethics committee or institutional review board? Not the police, right?

L69. Was instead of were

L72. Perhaps “minimize” is better than “avoid.”

L102. How, exactly, does the pharmacist confirm non-toxicity?

Author Response

(The authors gave the same response as above.)

Round 2

Reviewer 1 Report

The Revision of the manuscript Looks good to me. I feel all raised points were adequatly adressed.

There is only one : in the formula describing the conversion of Delta values realtive to PDB to atom-% 13C it seems to me that the "Delta" is missing

Author Response

(The authors gave the same response as above.)

Reviewer 2 Report

Our study suggests that DHA from AceDoPC may be more efficiently incorporated into RBC membranes…

42-3. Due to this low level of synthesis, dietary pre-formed DHA is the preferred source of DHA to improve brain DHA accretion.”

46ff. This is still not clear. LysoPC is “slower” it is “more efficient.” Faster but more limited for NE DHA?

second most abundant after DHA… where? In plasma, brain? “also is MORE efficient…”

60ff. at equivalent molar doses? Better said, "This compound affords more neuroprotection in a post-ischemic stroke model than does non-esterified DHA, and DHA incorporation into brain tissues is greater with AceDoPC than with equimolar concentrations of non-esterified DHA or of PC DHA." If this is true!

63-65. Move this statement to Discussion. It does not flow here.

130 g/what? taking this on a piece of bread with no other fats is a very important caveat that must be highlighted. Who knows how the results would have changed if taken in the more normal situation, with a meal.

Fig 1. I assume that the AUC difference is ns?

it is unclear why RBCs are even included in this experiment with only a single dose and such a short window follow up time. More relevant would be to focus only on plasma PL and look at PL subfractions. It need to be clarified why you studied RBC, and why you focused on RBC PC and PE but not plasma PC and PE. "transfer" cannot be "important"... it might be greater or faster, but not more important. if so then you should include an acknowledgement here that you don't know where the DHA molecule was on the TG you used. This is a limitation.

277,281. The use of terms like 'clearly' from a study with 3 people using a trademarked n-3 product from LipTher suggest a bias in favor of this potential pharmaceutical product instead of the unbiased opinion of a neutral observer. Do any of the authors have a financial interest in this product?

Author Response

I have now included some additional comments in the Discussion paragraph, with two additional references, in agreement with my answers to the Editor,

So, please find attached a text showing the different alterations, with the first revision in red color, the second revision in blue color, and the third revision (today) in green color.
